# Genome-Wide Novel Genic Microsatellite Marker Resource Development and Validation for Genetic Diversity and Population Structure Analysis of Banana

**DOI:** 10.3390/genes11121479

**Published:** 2020-12-09

**Authors:** Manosh Kumar Biswas, Mita Bagchi, Dhiman Biswas, Jennifer Ann Harikrishna, Yuxuan Liu, Chunyu Li, Ou Sheng, Christoph Mayer, Ganjun Yi, Guiming Deng

**Affiliations:** 1Institute of Fruit Tree Research, Guangdong Academy of Agricultural Sciences, Tianhe District, Guangzhou 510640, China; yuxuanliunxyc@163.com (Y.L.); lichunyu881@163.com (C.L.); shengou6@126.com (O.S.); yiganjun@vip.163.com (G.Y.); 2Department of Genetics, University of Leicester, Leicester LE1 7RH, UK; bagchi_econ@yahoo.com (M.B.); jennihari@um.edu.my (J.A.H.); 3The College of Economics and Managements, South China Agricultural University, Guangzhou 510640, China; 4Department of Computer Science and Engineering, Maulana Abul Kalam Azad University of Technology, West Bengal 700064, India; bdcse12@gmail.com; 5University of Malaya, Kuala Lumpur 50603, Malaysia; 6Forschungsmuseum Alexander Koenig, Bonn, Adenauerallee 160, 53113 Bonn, Germany; c.mayer.zfmk@uni-bonn.de

**Keywords:** *Musa* spp., comparative mapping, functional domain, SSR markers

## Abstract

Trait tagging through molecular markers is an important molecular breeding tool for crop improvement. SSR markers encoded by functionally relevant parts of a genome are well suited for this task because they may be directly related to traits. However, a limited number of these markers are known for *Musa* spp. Here, we report 35136 novel functionally relevant SSR markers (FRSMs). Among these, 17,561, 15,373 and 16,286 FRSMs were mapped in-silico to the genomes of *Musa acuminata*, *M. balbisiana* and *M. schizocarpa*, respectively. A set of 273 markers was validated using eight accessions of *Musa* spp., from which 259 markers (95%) produced a PCR product of the expected size and 203 (74%) were polymorphic. In-silico comparative mapping of FRSMs onto *Musa* and related species indicated sequence-based orthology and synteny relationships among the chromosomes of *Musa* and other plant species. Fifteen FRSMs were used to estimate the phylogenetic relationships among 50 banana accessions, and the results revealed that all banana accessions group into two major clusters according to their genomic background. Here, we report the first large-scale development and characterization of functionally relevant *Musa* SSR markers. We demonstrate their utility for germplasm characterization, genetic diversity studies, and comparative mapping in *Musa* spp. and other monocot species. The sequences for these novel markers are freely available via a searchable web interface called Musa Marker Database.

## 1. Introduction

Banana is one of the most consumed and commercially important fruit crops. Approximately 90% of the world’s bananas are produced in tropical and sub-tropical regions. Banana fruit significantly contributes to the export revenue and food security of these regions, and other banana plant parts are used locally for food and fibre. The annual global banana production in 2017–2019 was 116 million tons with an approximate value of USD 31 billion [1]. Most of the cultivated banana varieties are triploid, while some are diploid and have either a *Musa acuminata* (A) genome or are hybrids of the *Musa acuminata* and *Musa balbisiana* (B) genomes. The different levels of ploidy and chromosome numbers of banana have led to complexity in taxonomy [2] and are associated with parthenocarpy, leading to female sterility, seedless fruit and non-viable seeds. As a result, cultivated bananas are mainly reproduced asexually, with a consequent narrow genetic base. Genomic abundance, co-dominant inheritance, assay simplicity, and hyper variability make microsatellite markers (simple sequence repeats or “SSRs”) highly useful for genetic studies. The SSR markers located within protein-coding and the associated untranslated regions (UTRs) of genes may be directly linked with traits. Consequently, such functionally relevant SSR markers (FRSMs) can be useful to rapidly build marker–trait linkages and to discover genes/quantitative trait loci (QTLs) related to traits of agronomic importance in crop plants [3,4,5,6]. A number of studies have shown that functionally associated molecular markers are more powerful than anonymous markers for marker-assisted selection, marker-trait association [7], genetic diversity [4], comparative mapping [3] and the construction of transcript maps [8], and as anchor markers for evolutionary studies in plant species [3,4,5,8]. For banana, several hundred SSR markers have been developed from EST sequences [9,10,11,12], from BAC end sequences [13,14] and from A and B genome sequences [15]. However, only a small proportion of these markers have been experimentally validated and used for genotyping in banana. For example, Christelova et al. [16] used 19 SSR marker for genotyped 695 banana accessions. In addition, many of these banana SSR markers are not publicly available, and their chromosomal location, degree of polymorphism and functional characteristics have not been estimated which has limited the application of banana SSR markers in genotyping, association studies and fine mapping of important agronomic traits in banana breeding. Due to these limitations, there is a need for a well-curated SSR marker database based on information from in-silico data mining, including the genomic or chromosomic positions of SSRs, functional association and gene affinity, together with information based on experimental validation of SSRs, such as polymorphisms, transferability and allelic variations. This resource can accelerate various applications of genomics, genetics and breeding in bananas.

The application of molecular markers to banana breeding programs has included RAPD [17,18,19] (randomly amplified polymorphic DNA), ISSR [20,21,22,23] (inter simple sequence repeat), AFLP [24] (amplified fragment length polymorphism), DArT [25] (Diversity Arrays Technology), SSR [15] and SNP [26] (Single Nucleotide Polymorphism) and ASSP (Single Amino Acid Polymorphism) [27,28,29,30]. A recent review of the application of various molecular marker techniques in banana breeding showed RAPD, ISSR and AFLP marker techniques to be generally less effective compared to the SSR markers, due to low reproducibility, less allelic variation within single loci, and lack of co-dominance [31]. More recently developed NGS technology-based markers including RADseq, DARTseq and GBS provide high genome coverage and accuracy for genetic analysis compared to SSR markers. However, NGS approaches come at higher costs and require expertise for downstream application, while SSR genotyping is relatively simple assay and reasonable. An SSR genotyping approach is also able to systematically add new information to existing data sets. Moreover, SSR multi-allelic while SNP bi-allelic, so SSRs are more likely to detect polymorphism. As a result, banana breeders as well as other plant breeders still rely on SSR markers.

A large number of the transcript sequences from *Musa* spp. that are publicly available have not yet been mined for SSR markers. Recently, four whole genome sequences of the banana have been published and are available in the public domain. The total length of the *Musa acuminata* genome (A genome) sequence assembly [27,28] is 439 Mb representing 84% of the estimated size (523 Mb) of the DH-Pahang genome, which is distributed among 11 chromosomes. Similarly, a high quality 430 Mb (87%) draft genome of *M. balbisiana* [29] (B genome) was assembled into 11 chromosomes. The 525 Mb draft genome of *M. schizocarpa* [30] (S genome) was anchored to 11 groups, while the 462.1 Mb draft genome assembly of *M. itinerans* [31] covers 75.2% of the 615.2 Mb genome. The availability of these genome assemblies facilitates the extraction of geneic regions and the development and in-silico mapping of SSR markers. Transcriptome sequences are another valuable resource for identifying transcription factor (TF)-encoding gene products and TF-functional domains that are useful for developing functionally relevant molecular markers. The use of transcript sequences to identify SSR markers associated with TF-derived genes or functional domains has been reported in only a few plant species, including sugarcane [3], chickpea [32], tomato and pepper [33] but not previously for *Musa* spp.

In view of the above, the present study was conducted to (i) develop functionally relevant novel SSR markers from *Musa* transcript sequences and identify their loci, (ii) assess the cross-taxon transferability of developed markers, (iii) estimate the usefulness of functionally relevant Musa SSR markers for comparative mapping, (iv) evaluate the potential of these SSR markers for large-scale genotyping applications in *Musa* spp., (v) apply the SSRs in genetic diversity and population structure analysis of Musa germplasm and (vi) develop a searchable freely accessible SSR marker database.

## 2. Materials and Methods

### 2.1. Plant Material Collection and DNA Extraction

Eight genotypes (‘Prata’, ‘Kluai namwa khom’, ‘FHIA21’, ‘Dwarf Cavendish’, ‘FHIA 17’, ‘BITA2’, ‘BGY3’, ‘FHIA-03’) that represent major groups of banana and plantain (details of these listed in Appendix A) were used for the preliminary selection and transferability analysis of FRSMs (functionally relevant SSR markers). To estimate genetic diversity, population structure and phylogenetic relationships among *Musa* and related species (*Ensete* sp.), a total of 50 accessions were used. All the plant materials were collected from the core collection of *Musa* germplasm, maintained at FTRI, Guangzhou, PR China. Total genomic DNA was extracted from young fresh leaf samples using the CTAB method as previously described by Gawel and Jarret [34] with minor modifications.

### 2.2. Data Processing, SSR Mining, Marker Development, In-Silico Characterization, Physical Mapping and Functional Annotation

A total of ~0.1 Gb EST sequences of *Musa* spp. were acquired from public databases (EST Tool Kit and NCBI). All transcript sequences were combined into a single FASTA file with the aid of an in-house Perl script. After combining the sequences, the *est_trimmer.pl* (http://pgrc.ipk-gatersleben.de/misa/download/est_trimmer.pl) script was used to remove low-quality sequences, poly A/T and low-complexity regions at the 5′ and 3′ ends. The remaining high-quality sequences were assembled with CAP3 (http://mobyle.pasteur.fr/cgi-bin/portal.py#forms::cap3) with default parameters. Then mRNA sequences were extracted from genome assembly of recently published 4 musa genomes (A, B, S and I genomes) using perl script. SSR-containing mRNA (transcripts) and EST sequences were harvested using modified MISA script, then all the SSR-containing sequences were clustered with CDHIT using default parameters. The assembled non-redundant transcript sequences were searched for microsatellites using MISA (micro satellite identification tool, http://pgrc.ipk-gatersleben.de/misa/), restricting the output to perfect mono-, di-, tri-, tetra-, penta- hexa-, hepta-, octa-, nano- and deca- nucleotide motifs with a minimum of 10, 6, 5, 5, 5, 5, 4, 4, 4 and 4 repeat units, respectively. Identified SSRs were characterized based on (i) the length of repeat motifs (Class I > 20 bp, Class II ≤ 20 bp) and (ii) the nucleotide composition of repeat motifs (AT-rich, GC-rich and AT-GC balance). Forward and reverse primers for the identified SSRs were designed with primer3 software using default parameters. A custom Perl script called duplicate marker finder (http://mmdb.genomicsres.org/mumdb_Download.html) was used to remove redundant primer sets from the primer database. Further, non-redundant primer set compare with all the published Musa SSR markers and overlap markers were removed from the database. If more than one set of primers was generated for the same transcript sequence, one primer set was chosen randomly for further analysis. SSR-containing transcripts were analysed with the ORF (open reading frame) finder Perl script using default parameters to predict the longest ORF within the transcript. Finally, the localization of SSR motifs in either UTR or CDS/intron regions was determined by comparing the ORF position with the SSR position within the transcript sequence.

The genome assembly of *M. acuminata* (DH Pahang), *M. balbisiana* (Pisang Klutuk Wulung -PKW), *M. schizocarpa* and *M. itinerans* were downloaded from Musa Genome Hub (https://banana-genome-hub.southgreen.fr/species-list), and FRSMs were mapped onto the eleven chromosomes of these genome assemblies with the ePCR strategy. ePCR results were verified by BLAST searches of the markers against the whole-genomic sequences. Marker positions on the chromosomes were recorded and added to the database. A physical map was drawn using MapChart [35] software (Version 2.2). The specific in-silico-generated amplicons from these four genomes were compared with the expected amplicon size of each marker, and size differences were recorded. If an amplicon size differed by at least 2 bp, the SSRs were classified as polymorphic; otherwise, amplicons were considered monomorphic.

The putative function of each FRSM was determined by Blast2GO analyses. Flanking regions (200 bp up and 200 bp downstream) of the FRSM loci were searched against the non-redundant protein database in GenBank using BLASTX [36] with an E-value threshold of 10^−10^. Based on the functional annotation results, FRSMs were classified according to three GO terms: biological process, molecular function and cellular process [37,38,39]. The KEGG database (http://www.genome.jp/kegg/pathway.html) was used to annotate the pathways for these FRSMs. To identify transcription factor (TF)-associated and TF-functional domain-associated FRSM markers, we conducted a BLAST search of SSRs against the plant TF database (http://planttfdb.cbi.pku.edu.cn/) with an E-value threshold of 0.001 and a query coverage of 65%.

### 2.3. In-Silico Comparative Genome Mapping

The flanking sequences of FRSM loci were searched with BLAST against the genome sequences of *M. acuminata*, *M. balbisiana*, *M. schizocarpa*, sorghum, foxtail millet and rice to obtain marker-based syntenic relationships among the species. A bit score threshold of 54.7 and E-value threshold of 10^−5^ were considered as significant for this BLAST analysis. To check the presences of SSRs in the orthologous region of the target species, we chose the flanking sequences from orthologous region of the target species, found the SSRs using the MISA script and compared the repeat patterns and lengths with their corresponding FRSM loci. The marker-based syntenic relationships were finally visualized with Circos 0.55 (http://circos.ca).

### 2.4. Evaluation of PCR Amplification and Genetic Marker Potential

To assess the amplification efficiency of the in-silico-developed FRSM primer, a subset of 273 primer pairs was selected based on the in-silico results including, transferability, polymorphism, known gene functions and the physical distance between two primer sets should be 1.5 to 2 mb. The primers were synthesized by Sangon Biotech Co., Ltd., Shanghai, China; PCR amplifications were carried out for eight *Musa* accessions (Appendix A) representing diverse genomic groups of the core collection of the *Musa* germplasm, maintained at FTRI, Guangzhou, PR China. The FRSM primers displaying clear and reproducible amplicons in a gel-based assay were selected for the genotyping of 50 accessions of the *Musa* germplasm collection (Appendix A). PCR amplification was carried out in 25 µL containing the following: 50 ng of genomic DNA, 1 μL of each primer pair (concentration, 10 mM), 12.5 µL of PCR-Mix (Takara), 1.0 U of Taq DNA polymerase enzyme and 8.5 µL of dH_2_O. PCR amplification was carried out in a MJ-PTC-200 tm thermal controller (MJ Research, Waltham Mass) using the following programme: 94 °C for 3 min, 35 cycles at 94 °C for 30 s, 55–60 °C (according to the primer annealing temperature) for 30 s, 72 °C for 45 s, followed by a final step at 72 °C for 7 min. The amplified PCR products were resolved on a 3% agarose gel, and the band size was measured with the aid of a 100 bp DNA ladder. PCR products of selected primer pairs were resolved on a denaturing 6% polyacrylamide gel, and bands were visualized by silver staining.

### 2.5. Data Collection and Statistical Analysis

The PCR amplification result was recorded for each marker. The SSR bands of all genotypes were scored as band present (1) or absent (0). Allele frequencies were used to calculate the polymorphism information content (PIC) using the following formula: $PIC=\;\sum p_{i}^{2}$, where *p_i_* is the proportion of the *i*th allele. PIC values and allele frequencies were calculated using Power Marker software. The NEI72 module (genetic distance data module) of the NTSYSpc software package (version 2.1 [40]) was used to estimate genetic similarity among genotypes. Based on the NEI72 similarity matrix, UPGMA dendrograms were constructed using the SAHN module. The FIND module was employed to identify all trees that could result from tied similarity values. Mantel test statistics were used to assess the reliability of the clustering by comparing the cophenetic matrix [40] and the similarity matrix. The effective number of alleles (*ne*), Shannon’s information index (*I*), expected homozygosity (*Ho*), expected heterozygosity (*He*), Nei’s expected heterozygosity, Wright’s fixation index (*Fis*) and the Ewens-Watterson Test for Neutrality were calculated using Popgene software [41].

### 2.6. Population Structure

The population structure of 50 banana accessions was estimated using STRUCTURE 2.3.4 software [42]. The admixture model was used with a burn-in period length of 10,000 and 100,000 MCMC iterations. Five independent runs were performed for each k from 1 to 10. The best value of k was estimated based on the Delta k method using Structure Harvester software [43]. Barplots of the Q matrix were drawn using DISTRUCT software [44]. Principal coordinate analysis (PCoA) for population separation was carried out using the dissimilarity matrix data of each individual (genotype) and using GenAlEx software [45].

### 2.7. Functional SSR Marker Database

In order to maximize the utility and ensure the availability of developed novel functional Musa SSR markers, here we developed a searchable database using HTML, CSS, JavaScript, PHP and MySQL based coding language. PHP script was used to bridge the search interface and database. JavaScript, PHP and HTML use for results visualization, and download in XLS or CSV format. First all the marker information (29 attributes) were organized in a table with a unique marker id. Then all these makers with their corresponding attributes stored in a mySQL database. The database and database search interface hosted in the published web site called Musa marker database (http://mmdb.genomicsres.org/index.html).

## 3. Results and Discussion

### 3.1. FRSM Marker Development, In-Silico Characterization, Chromosomal Distribution, Physical Mapping and Genome Coverage

A total of 68,716 SSR-containing transcript sequences were extracted using MISA (with perfect mono-, di-, tri-, tetra-, penta- hexa-, hepta-, octa-, nano- or deca-nucleotide motifs with a minimum of 10, 6, 5, 5, 5, 5, 4, 4, 4 and 4 repeat units, respectively) from 222,124 transcripts sequences that were originally obtained from four published Musa genome sequences and ~0.1 Gb EST sequences (Appendix A). Subsequently those SSR-containing transcripts were assembled into 52,453 unigene sequences and searched for simple sequence repeats (SSRs) to design functionally relevant SSR markers for *Musa*. The type and frequency distribution of SSR motifs are presented in Table 1. The proportion of SSR-bearing *Musa* unigenes was 31% (68,716 out of 222,124), which is higher than those reported for coffee, cassava, and cereals [46,47,48,49] and lower than those reported for iris and sugarcane [3,50]. The frequency of SSRs in *Musa* unigenes was one per 2.2 kb, which is higher than that reported for sugarcane transcript SSRs (1SSR/10.9 kb) [3], but similar to reports for rice (1SSR/3.6 kb), sorghum (1SSR/5.9 kb) and barley (1SSR/8.9 kb) [51,52]. The proportion of SSR-bearing unigenes was much higher in this study compared to earlier observations based on EST sequences of *Musa* [53]. This difference may be due to the variation in search criteria, size of data sets and software tools used in each study, as suggested previously for similar comparisons [5,54,55,56,57,58,59]. Out of 51,814 SSR-bearing unigenes, 30,780 (59.4%) contained more than one SSR motif. Among the different repeat units, di-nucleotide (51.1%), mono-nucleotide (31.5%) and tri-nucleotide (14.6%) repeat motifs were more frequent than other repeat motifs. In addition, tri-nucleotide repeats were more frequent in CDS (coding sequences) regions (see Appendix A). The same trend was also reported for *Musa* EST-SSRs and SSRs in cereals and legumes [51,60]. The high abundance of tri-nucleotide repeat motifs in the coding region of most genomes can be explained by the fact that length-altering mutations do not change the reading frame. Based on the repeat motif length, 59.6% SSRs were characterized as Class II type (12–20 bp), and the remaining 40.4% were classified as Class I type (>20 bp). Backiyarani et al. [9] also found more class II SSRs than class I SSRs in *Musa* EST sequences. From 113,480 identified SSRs (di to hexa repeat SSR use for marker development), reverse and forward primers could be designed for 80,899 (71.3%) SSRs. For the remaining 28.7%, primer design failed due to insufficient flanking regions or due to insufficient sequencing quality in the genomic region of interest. Similar results have been reported for many other plant species [57,61,62]. Primer redundancy is a major problem in SSR primer design projects, especially if a large number of primers are designed for a large set of sequences [63]. Primer redundancy may be the consequence of chromosomal or gene duplications. Alternatively, some of the EST sequences possess multiple SSR tracts, in which case multiple primer sets might be generated from the same flanking sequences. To filter redundant and multiple sets of primers from the same flanking sequences, a Perl script called *Duplicate Marker Finder* (http://mmdb.genomicsres.org/mumdb_Download.html) was used to reduce the SSR primer set number to 66,246. In order to develop novel SSR markers, non-redundant markers were compiled with the published Musa SSR markers and 35,136 (43.4%) novel Musa SSR markers were found. These primer pairs were then characterized by both in-silico and wet lab methods. Out of the 35,136 primer pairs, 1104 (1.7%) were characterized as transcription factor-associated primers (Table 1).

Among the TF-associated FRSM, 11%, 8% and 7% were related to the transcriptome factor families ERF, NAC and C2H2 (Figure 1). In total, 13,223 (37.6%) primers were found to originate from CDS, 11,718 (33.4%) from 5′ UTR and 10,099 (28.7%) from 3′ UTR regions. Expansion or contraction of SSR repeats in the 5′ UTR region of a gene may affect transcription and/or translation, while SSR repeat unit variation in the 3′ UTR region may be responsible for gene silencing or transcriptional slippage. SSR repeat polymorphisms in CDS regions may lead to the activation or deactivation of gene function by truncating or extending proteins [5]. Therefore, SSR markers derived from different regions (CDS, 5′UTR and 3′UTR) of genes may have different characteristics. Scott et al. [64] observed polymorphism differences among SSR markers derived from different regions of genes in the grape genome. The 5′ UTR-derived SSR markers are mostly polymorphic between cultivar and species, while the 3′ UTR-derived SSR markers are polymorphic at the cultivar level and CDS-derived SSR markers are polymorphic between species and genera. In terms of polymorphism detection, 3′-UTR-derived SSR markers are superior to those derived from 5′-UTRs [4] for cultivar identification and genetic diversity analysis. In view of the association of SSR from different gene regions with variation at different taxonomic levels, from previous studies [5,64], we propose that SSR markers derived from different regions of *Musa* genes should be used for different applications in banana breeding programmes; for example, 3′UTR-derived *Musa* SSR markers may be most suitable to distinguish banana cultivars.

The genomic distribution and physical localization of the 35,136 SSR markers on the A, B and S genomes of banana are presented in Table 2 and in Appendix A. A total of 17,561 (50.0%), 15,373 (43.8%) and 16,286 (46.4%) markers mapped on the eleven chromosomes of the genomes of *M. acuminata* (A genome), *M. balbisiana* (B genome) and *M. schizocarpa* (S genome), respectively. All physically mapped markers are available in the *Musa* Marker Database (http://mmdb.genomicsres.org/mumdb_Download.html). The chromosome-wide distribution and frequency of the physically mapped markers revealed the highest frequency on chromosome 4 for the A and S genomes (2203 markers (6.3%) for the A genome; 2080 markers (5.9%) for the S genome), while for the B genome, the highest number of markers mapped on chromosome 8 and the lowest frequency on chromosome 1 (1176 markers (3.3%) for the A genome; 1227 markers (3.5%) for the S genome). The highest and lowest marker densities were found on chromosome 4 and chromosome 9, respectively, in both the A and S genomes. The physical map constructed in this study is highly dense compared to the physical map reported for foxtail millet [62]. The high-density SSR marker-based physical map constructed in this study will be helpful for the selection of suitable genome-wide SSR markers across the eleven chromosomes for various applications in improved banana breeding programmes, including large-scale genotyping, comparative genome mapping, QTL analysis and population genetic studies.

### 3.2. Functional Annotation and Association of Transcription Factor

To identify the functional significance of the 35,136 novel microsatellites markers, a BLASTX analysis was performed against the non-redundant protein database using the Blas2Go tools. A total of 14,312 (41%) were functionally annotated due to their high similarity (e value < e^−10^) with at least one protein in the non-redundant protein database (Appendix A). The remaining 20,824 (59%) did not show a significant similarity to known protein sequences in the databases and therefore were not annotated. According to the Gene Ontology (GO) scheme, functionally annotated FRSMs were classified into three categories: cellular components (CC), molecular function (MF) and biological process (BP) (Appendix A). The sum of these FRSMs per category did not add up to 100% because some FRSMs were classified into more than one category. A total of 12,120 FRSMs were annotated with 44,279 annotations for the three main GO categories, while 3891FRSMs were annotated with all three GO categories and 4148 were assigned at least two GO annotation categories (Appendix A). Among the three categories, BP annotations were most frequent, followed by CC and MF (Appendix A). SSR-bearing transcript with CC annotations were mainly annotated as intracellular organelle (26%), cytoplasm (22%), membrane (21%) and cell periphery (5%) (Appendix A). Under the MF category, most of the SSR-bearing transcripts were annotated as organic cyclic compound binding (26%), followed by heterocyclic compound binding (26%), transferase activity (16%), and hydrolase activity (13%) (Appendix A). Among the SSR-bearing transcripts with BP annotations, annotations suggested that the proteins are involved in primary metabolic process (20%), nitrogen compound metabolic process (17%), cellular metabolic process (16%) and biosynthetic process (11%) (Appendix A).

### 3.3. In-Silico Comparative Genomic Mapping between Musa and Non-Musa spp.

Comparative mapping of the FRSMs among the three Musa genomes is illustrated in Figure 2a–d and revealed that 5938 (16.9%) of the FRSMs were common among the four Musa genomes, while only 11 (0.03%) were found to be common in the genomes of four Musa and four non-Musa species (Figure 2a,b). A total of 17,403 FRSMs physically mapped onto the eleven chromosomes of *M. acuminata.* We compared their physical location on the *Musa* genome with their location on the chromosomes of the related monocot plant genomes of foxtail millet, rice and sorghum (Figure 2c,d, Table 3). In-silico comparative genomic mapping showed a considerable proportion of sequence-based orthology and syntenic relationships, with SSR markers distributed over eleven *Musa* chromosomes with the chromosomes of foxtail millet (~1%, 240), rice (~1%, 264) and sorghum (~2%, 268) (Figure 2, Appendix A). Many of these markers show syntenic relationships with more than one chromosome of foxtail millet, sorghum and rice. This result hints at segmental duplication events among the chromosomes of the respective genomes, similar to the observations reported by Pandey et al. [62], that many of the physically mapped foxtail millet SSR markers have syntenic relationships with more than one chromosome of sorghum, maize and rice.

In-silico comparative mapping between *Musa* and foxtail millet genomes demonstrated a syntenic relationship of 240 SSR marker loci distributed on eleven chromosomes of the *Musa* genome with 480 genomic regions on 9 chromosomes of the foxtail millet genome (Figure 2d; Appendix A). The higher frequency of SSR marker-based syntenic relationships with foxtail millet and Musa was (10; 42%) found between Foxtail millet chromosome 3 and Musa chromosome 1 (Table 3). A total of 264 SSR marker loci that physically mapped onto the eleven chromosomes of *Musa* showed synteny with the 12 chromosomes of rice with an average frequency of 9%, which is lower than what we found for the foxtail millet and sorghum chromosomes (Figure 2d; Appendix A). The 264 *Musa* SSR marker loci showed significant matches with 436 genomic regions of 12 rice chromosomes. Overall, rice chromosome 1 showed a high frequency of syntenic relationships with most of the *Musa* chromosomes, except Chromosomes 1, 4 and 6. The syntenic relationship between the *Musa* and sorghum genomes revealed that 268 SSR marker loci have significant matches with 656 genomic regions on 10 chromosomes of sorghum. Maximum syntenic relations were found between *Musa* chromosome 1 and sorghum chromosome 1 (11, 41%) (Figure 3; Appendix A).

In-silico mapping often retains the orthologous region of the target species without SSR regions, and this phenomenon may limit the applicability of SSR markers in other species, especially distant relatives. Therefore, we checked all orthologous regions from the target species for the presence of SSR repeats. Our results showed that 83.6% (for *M. acuminata*), 93.2% (for *M. balbisiana*) and 91.9% (*M. schizocarpa*) of the homologs (Appendix A) may have either the same or shorter SSR lengths than the reference SSR motifs. However, a small percent of the orthologous regions of the foxtail, rice and sorghum genomes contain SSR motifs. These findings indicated that homologous regions may exist in the targeted species, but some of them lacked SSR motifs.

Although many studies have reported comparative mapping analyses of plants based on SSR markers [62,65,66,67,68], this is the first report of an SSR marker-based comparative genome mapping analysis of *Musa* with foxtail millet, rice and sorghum. SSR marker-based comparative genome mapping between Musa and other non-Musa species could allow the transfer of marker information among these target plant species, consequently accelerating map-based gene isolation of important agronomic traits in *Musa* species. In this study, we recorded the highest syntenic relationships between *Musa* and sorghum, followed by between Musa and foxtail millet, and between Musa and rice. A lower degree of syntenic relationships was recorded in this study compared with the previous reports of Pandey et al. [62] in which comparative maps were constructed between foxtail millet and sorghum as well as between foxtail millet and maize. As foxtail millet, sorghum and maize species belong to the same subfamily, *Panicoideae*, these are more closely related than the species compared in our current study. The similarities found among the genomes are in line with their evolutionary distances and conform to the expectation that syntenic relationships among plant species decrease with increasing phylogenetic distance [62,66]. Our findings show that microsatellite marker-based comparative genome mapping of *Musa* and other plant species (monocots) provides a first picture of genome conservation among Musa, foxtail millet, rice and sorghum. Furthermore, these comparative maps will be useful in map-based gene cloning from Musa, which is agronomically important for obtaining marker-based genotyping information from other related plant species.

### 3.4. PCR Amplification Efficiency, Polymorphism, Transferability and Genetic Marker Potential

A sub-set of 273 selected FRSMs was tested for PCR amplification efficiency, polymorphism, transferability and genetic marker potential (Figure 3; Appendix A). Among these markers, 259 produced a clear, reproducible PCR amplification product of the expected size. A total of 249 (91%) markers produced single bands, whereas 24 markers had more than one band. Furthermore, 36 markers produced either longer or shorter fragments than the expected size. Overall, 203 (74%) of the markers showed polymorphism in the wet lab (PCR) assay among eight accessions of *Musa*, and a total of 715 different alleles were detected among the eight accessions. The number of alleles per marker ranged from 2 to 12 (Appendix A), with an average of 3.5 alleles. Trinucleotide repeat motif-containing markers showed the highest percentage of polymorphism both from in-silico and wet lab assays (Figure 3). It should be noted that the proportions of markers found to be monomorphic and polymorphic in the in-silico and in the wet lab assays differed substantially, most likely because a larger number of accessions (Appendix A) were used for the PCR experiments, while genomic sequences were not available for all accessions for the in-silico polymorphism assay. Therefore, only three (A, B and S genomes) available genomes were used for the in-silico polymorphism assay.

Transferability to closely related species is one of the most important potential features of SSR markers, offering the opportunity to use markers developed from one species to be applied to multiple species. This is particularly useful if little is known about the genomic sequence of the species of interest. Transferable markers are a good source for detecting orthologous loci between two species and are useful for incorporation into genetic maps. To estimate the cross-species transferability of the markers developed in this study, a set of 273 markers were analysed for the ability to amplify genomic DNA of eight banana species representing different genomic groups (Appendix A). A total of 194 (71%) markers produced amplicons in at least three species among the eight *Musa* species. From the literature, it was found that few markers are transferable to less closely related species; for example, on average, 25% of the *Centella asiatica* EST-SSR markers were shown to be transferable to the family Apiaceae, while 40% of the barley EST-SSR markers were found to be transferable to rice [46,69]. Higher levels of SSR marker transferability have been reported for other plant species, including citrus, millet and sugarcane [57,62,66]. The transferability of the SSR markers between species or families greatly relies on the evolutionary distance between the species [66]. In this study, we tested the SSR marker transferability between *Musa* species (*M. acuminata* and *M. balbisiana* and between hybrids of the two) while the above-mentioned studies by Thiel et al. [46]. Sahu et al. [69] examined marker transferability at the genus and family levels. Consequently, our marker transferability rate is expected to be higher than that found by Thiel et al. [46] and Sahu et al. [69]. The high transferability rate of FRSMs between banana species supports their utility in comparative mapping studies and for the identification of markers associated with important agronomic traits.

### 3.5. Assessment of Genetic Diversity and Population Structure

One of the most important applications of SSR markers is the assessment of genetic diversity within natural populations, among core collections of germplasm and among breeding lines. Such genetic variation information is vital for efficient conservation and for the use of genetic resources for crop improvement programmes. Consequently, a core set of 50 Musa accessions and 15 FRSMs was used to determine the effectiveness of the selected FRSMs for the assessment of genetic diversity in *Musa* spp. In total, 49 alleles were identified in 15 loci with an average of 3.3 alleles per locus (Appendix A). The PIC (polymorphic information content) values for individual loci ranged from 0.26 to 0.66 with a mean of 0.50 and the effective number of alleles per locus varied from 2.02 to 3.44. The Shannon information index (I) ranged from 0.74 to 1.44, with a mean of 0.94. The average PIC value found in this study was comparable with that reported for SSR markers in citrus species by Biswas et al. [57] and higher than that reported for SSR markers in *Musa* by de Jesus et al. [70]. As PIC values are greatly influenced by the number of genotypes and genetic background of the genotypes, this explains the discrepancy between the current study and the findings reported by de Jesus et al. [70], which used different sets of germplasm. The average observed and expected heterozygosity were 0.71 and 0.62, respectively (Appendix A). The Nei’s average genetic diversity [71] ranged from 0.51 to 0.71, with an average of 0.62. The fixation index (*F_IS_*) was found to be positive for four loci (C01P3AA00134, A2M000167, AB2M006866 and NovelTSSR001634) and negative for the remaining eleven loci (Appendix A). In this collection of loci, the majority were homozygous in the tested population and thus should be informative for population genetic studies in banana. The overall mean value of the fixation index was 0.17, revealing an excess of heterozygosity present in the studied population. Several factors can result in an excess of heterozygosity in plant populations, including a wide range of cross-compatibility and polyploidy and a high proportion of loci under selection and allele-scoring biases. Most likely, the excess heterozygosity observed in this study is due to the polyploidy of the banana accessions studied, which are representative of commonly grown cultivars. Since population genetics applications require selectively neutral markers to avoid misleading results [72], neutrality tests have been advocated, for example by Kim et al. [73]. Therefore, in this study, we performed the Ewens-Watterson test for the neutrality of 15 markers and observed frequency values ranging from 0.291 to 0.494 (Appendix A). At all loci, the mean expected homozygosity value was higher than the observed value.

The clustering of subpopulation based on k-value (Figure 4a), UPGMA tree (4b) and admixture model base population structure was assed to test the strength of the FRSMs. Phylogeney analysis showed that 50 banana accessions (Figure 4b) clustered into two major groups according to their genomic background, with cluster II including all genotypes with only the A genome, while the AB genomic composition genotypes grouped in cluster I. According to the genome composition and ploidy level, cluster I and cluster II were further divided into two and five sub-groups, respectively. Most of the accessions of sub-group 1 were diploids and triploids. Sub-group 2 was composed of diploids and triploids; while sub-group 3 included most of the accessions from the Musa sub-group Mutika/Lujugira and in this sub-group the members were diploid, triploid and tetraploid. In sub-group4, most of the members were triploid Cavendish banana. As expected *Ensete* sp. was located in a distinct clade outside of the *Musa* species. Cluster analysis clearly showed that diploid A accessions share sub-clusters with other triploid and tetraploid accessions of the A genomic group. Similar results were also reported by earlier studies [74]. These observations may suggest that the variation within subspecies of *M. acuminata* is complex and that cultivars of *M. acuminata* have a wide partition from their fertile diploids rather than their cultivated AAA relatives. In this study we noticed that most of studied genotypes were different from each other with few exceptions. For example, we cannot distinguish genotype “Nyamwihogora” and “Intokatoke”, both are triploid and composed with A genome. Our SSR marker was unable to differentiate “Igitsiri” and “Ingagara”. We were also unable to distinguish two cavendish cultivars “Dwarf Parfitt” and “Formosana”. This finding is due to the genomic similarity of these genotypes. Such as triploid A genome type cultivars like cavendish are known to be less diverse cultivars. Different types of molecular markers such as PCR-RFLP, SSR and ITS failed to distinguish completely all the triploid A genome type cultivar [74]. Considering our findings and previous evidence, we recommended SNP or DART types of molecular markers may be suitable for distinguishing triploid A genome type cultivars of banana. To verify the adjustment between similarity matrices and respective dendrogram-derived matrices, we estimated the cophenetic correlation coefficient (r), and the significance of the correlation was determined by the Mantel matrix correspondence test. The correlation coefficient was statistically significant (Appendix A), demonstrating that the FRSMs are capable of distinguishing the A and AB genotypes of *Musa* spp.

A Bayesian clustering model-based analysis estimated the distribution of 49 alleles at 15 SSR loci among 50 accessions of banana. The number of subpopulations (k) was estimated to range between 2 and 10. To identify the approximate number of subpopulations, we estimated the maximum value of the logarithm of likelihood (LnP(k)). However, we did not find the LnP(k) value to achieve a clear plateau, and the value continued to increase together with the variances between the tested k. Under these conditions, the k value was anticipated to be between 5 and 7 (Figure 4a). We noticed that there was no longer variation for the population grouping at the k values of 5, 6 and 7 (Figure 4a). In this situation, the highest peaked value of k is more suitable to estimate the number of subpopulations than the lowest peak value [75]. This approach fit well to identify the sub-population in banana [70]. Therefore, we adopted this approach for identifying the number of sub-populations, and the highest k value (k = 7) was chosen for 50 banana accessions distributed in six subpopulations. This subpopulation clustering clarifies the underlying genetic structure among the studied banana accessions. A Principal Coordinates analysis (PCoA) estimate of the efficiency of tested FRSMs grouped the 50 banana accessions into two major categories according their genome composition (Appendix A). Therefore, PCoA analyses are also able to differentiate banana accessions according to the main genotypes. Phylogenetic clustering methods for detecting genomic compositions were also reported for bananas by Ning et al. [74]. Our STRUCTURE analysis demonstrated clear separation of the 50 banana germplasm accessions according to their genomic background and provided further evidence of the utility of our FRSMs for the characterization of banana germplasm.

Altogether, we were able to show that FRSMs are promising markers for successful large-scale genotyping analyses, germplasm characterization, genetic diversity, and population structure studies in *Musa*.

### 3.6. Novel Functional Musa SSR (NFMS) Marker Database Developments

All the developed (35,136) novel functional Musa SSR markers with 29 features have been placed in the freely search able marker database. This marker database contains eight flexible search fields including SSR Type, SSR Class, SSR Motif rich with, SSR Polymorphism, SSR Transferable to other plant species and SSR associated with gene function (Figure 5a–c). The search results display as a list of markers with Marker ID, forward- and reverse-primer sequences, and the sequence id, with links to additional information including all the 29 features such as transferability, polymorphism, genome position, flanking sequences and functional annotations. Query results can be downloaded in XLS and CSV file formats for subsequent use.

Identifying the most suitable SSR markers from a large data set is both challenging and time consuming. The availability of a well-structured and searchable SSR marker database facilitates the selection of desirable SSR markers for breeders, as well as reducing the time, cost and labour. Hence, we built the banana SSR marker database to store and to share the novel functional *Musa* SSR markers identified by our study. SSR marker databases have been developed for several plant species including rice [76], maize, sorghum, soybean [77], tomato [78], chickpea [79], pigeon pea [80], and also previously for banana [81]. The existing online genomic resources for Banana including the Banana Genome Hub (http://banana-genome-hub.southgreen.fr/home1) (BGH), TropGENE Database (http://tropgenedb.cirad.fr/tropgene/JSP/index.jsp) and BanSatDb database (http://webtom.cabgrid.res.in/bansatdb/index.php) also contain molecular marker information Among these three databases, BGH and TropGENE store only Musa SNP markers, while BanSatDb was developed for Musa SSR markers. BanSatDb allows the design SSR primers from the genomes of three *Musa* species: *M. acuminata*, *M. balbisiana*, and *M. itinerans*, but not for *M. schizocarpa.* There are some other important limitations of the BanSatDb database including the very limited and inflexible search parameters: It does not permit a user to develop primers from the genome; it cannot provide the primer redundancy, transferability, or information on polymorphism to the other Musa and non Musa species; and it does not provide SSR motif information in the designed primers. Overall, there is no unique identifier for the designed primers which can affect reproducibility in downstream application of the markers by the same or different researchers. The NFMS data base, by comparison, is more user friendly, flexible and contains a much larger volume of information for each developed marker: Each maker entry has 29 different features to assist users to easily select the most suitable marker pair from their query result. Users are able to download the entire search result in a single XLS or CVS format file within a single click. These unique features, the volume of the information and the quality of data in the NFMS database provide a user friendly, unique and versatile novel Musa marker database.

## 4. Conclusions

Here, we present the first set of functional SSR markers (FRSMs) developed for *Musa* spp. together with a freely searchable database. A large number of FRSMs were mapped onto the *Musa* A, B and S genomes. The markers were characterized by both in-silico and wet lab techniques, and their utility for genetic diversity studies was estimated. Our results demonstrate the utility of FRSMs for large-scale banana germplasm characterization, comparative mapping and genomic-associated studies among *Musa* and non-*Musa* spp.

## Figures and Tables

**Figure 1 genes-11-01479-f001:**
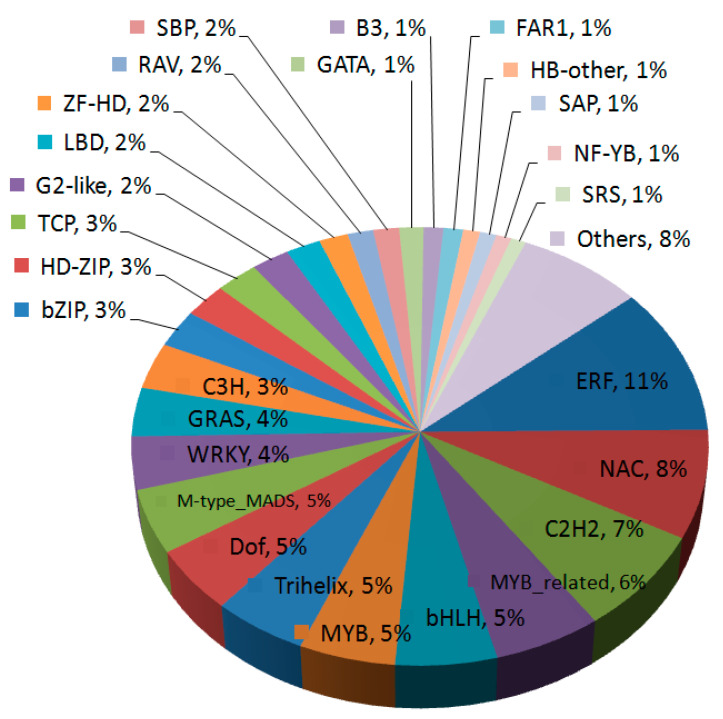
Functional annotation of transcript sequences carrying microsatellites in their functional domains. A total of 1104 SSR-bearing transcript sequences were associated with functional domains of TF.

**Figure 2 genes-11-01479-f002:**
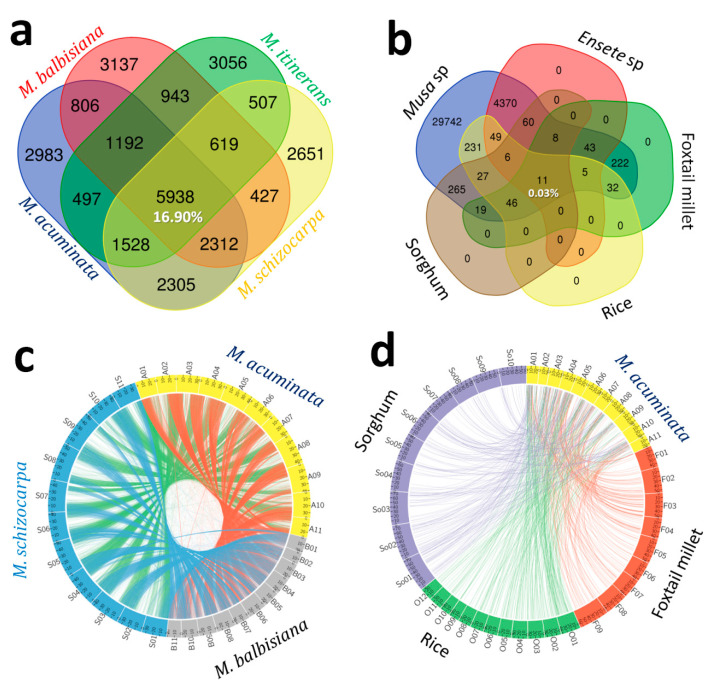
Functionally related microsatellite marker-based genomic relationships of Musa with other plant species. (**a**) Venn diagram representing the common FRSM markers among the three Musa genomes. (**b**) Venn diagram representing the common (transferable) FRSM markers among the Musa and non-musa genomes. (**c**) FRSM-based syntenic relationship among the three *Musa* genomes. (**d**) FRSM-based syntenic relationship among the *Musa* and non-*Musa* genomes.

**Figure 3 genes-11-01479-f003:**
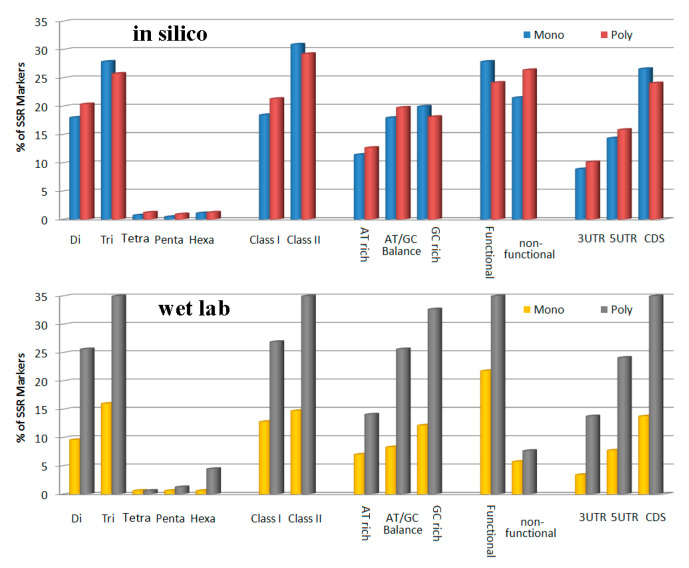
Polymorphic and monomorphic marker distribution among different classes based on in-silico and wet lab analyses.

**Figure 4 genes-11-01479-f004:**
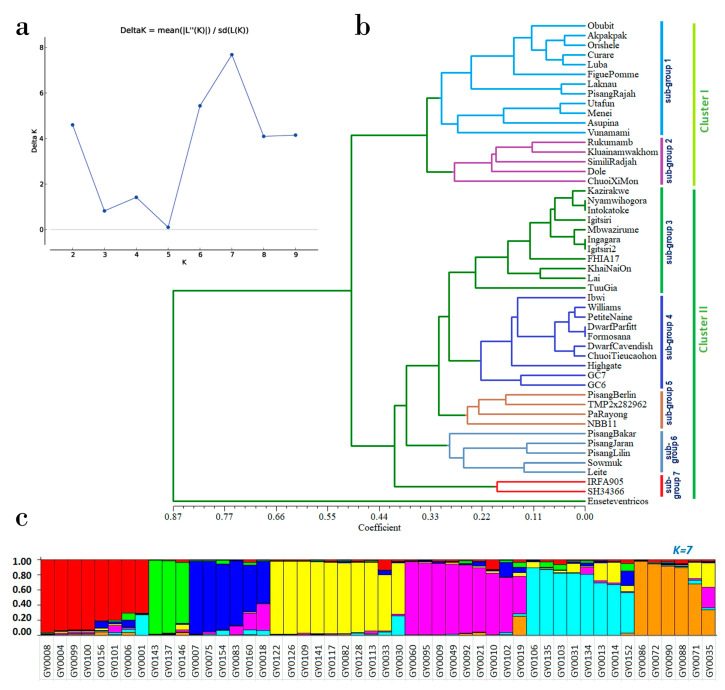
Phylogenetic relationship and population structure of 50 *Musa* accessions using 15 FRSMs; (**a**) plot of delta K values for different numbers of population assumed; (**b**) UPGMA dendrograms constructed based on the NEI72 similarity matrix; (**c**) sub-populations represented by distinct colours. Each column represents each accession (the name of the accession are presented in Appendix A) that can be fractionated into segments, and the whole size has the potential to estimate membership fractions (q) in k clusters. The genomic background of each accession was based on morphological descriptions recorded in the Musa germplasm database.

**Figure 5 genes-11-01479-f005:**
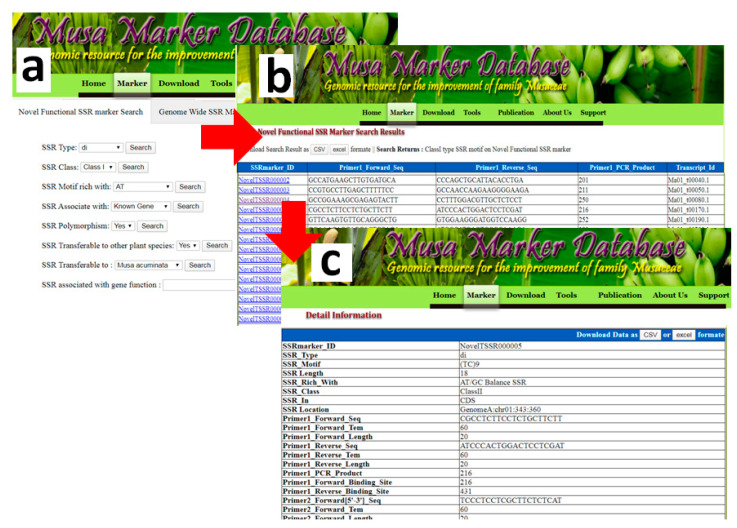
The flow of a database search in NFMS. (**a**) marker search interface (**b**) query result (**c**) details information of each marker.

**Table 1 genes-11-01479-t001:** Distribution of microsatellites in transcript sequences and functionally relevant SSR marker development.

Characters under Study	Unigenes	%
Total number of sequences examined	52,453	
Total size of examined sequences (bp)	359,005,912	
Total number of identified SSRs	166,503	
Number of SSR-containing sequences	51,814	
Number of sequences containing > 1 SSR	30,780	
Number of SSRs present in compound form	47,796	
SSR density (1 SSR per bp)	2156	
Class II SSRs (12–20 nucleotides)	99,231	59.60
Class I SSRs (>20 nucleotides)	67,272	40.40
AT rich SSRs	112,814	67.75
GC rich SSRs	17,073	10.25
AT/GC balance	36,616	21.99
Mononucleotide repeats	52,440	31.49
Dinucleotide repeats	85,090	51.10
Trinucleotide repeats	24,330	14.61
Tetranucleotide repeats	2884	1.73
Pentanucleotide repeats	413	0.25
Hexanucleotide repeats	763	0.46
Heptanucleotide repeats	180	0.11
Octanucleotide repeats	238	0.14
Nanonucleotide repeats	28	0.02
Decanucleotide repeats	137	0.08
No. of primers designed	80,899	71.29
Non-redundant primers	66,246	81.89
Novel SSR primer identified	35,136	43.43
TF-associated SSRs	1104	1.67
Transferable to other *Musa* sp.	18,209	51.82
Genome specific	16,927	48.18
In-silico polymorphic	8605	24.49
No. of SSRs in CDS	13,223	37.63
No. of SSRs in 5′ UTR	11,718	33.35
No. of SSRs in 3′ UTR	10,099	28.74

**Table 2 genes-11-01479-t002:** Summary of chromosomal distribution and average density of SSR markers mapped on the genomes of *M. acuminate* (A genome), *M. balbisiana* (B genome) and *M. schizocarpa* (S genome).

Genome	No of Markers Mapped						Density (1 Primer/Mb)		
	A	(%)	B	(%)	S	(%)	A	B	S
Chr1	1176	3.3	1227	3.5	1098	3.1	24.72	17.96	32.80
Chr2	1215	3.5	1276	3.6	1151	3.3	24.29	13.60	32.35
Chr3	1731	4.9	1155	3.3	1639	4.7	20.23	20.92	27.33
Chr4	2203	6.3	1001	2.8	2080	5.9	16.84	24.63	23.67
Chr5	1566	4.5	1512	4.3	1478	4.2	26.73	15.64	35.19
Chr6	1982	5.6	1547	4.4	1649	4.7	18.97	17.99	26.06
Chr7	1464	4.2	1276	3.6	1377	3.9	23.93	17.41	34.13
Chr8	1824	5.2	1567	4.5	1696	4.8	24.61	17.66	32.35
Chr9	1486	4.2	1232	3.5	1428	4.1	27.80	21.02	36.93
Chr10	1498	4.3	1564	4.5	1443	4.1	25.15	16.13	30.01
Chr11	1258	3.6	1204	3.4	1170	3.3	22.22	17.21	31.42
ChrUn	158	0.4	812	2.3	77	0.2			
Total Mapped	17,561	50.0	15,373	43.8	16,286	46.4			
Unmapped	17,575	50.0	19,763	56.2	18,850	53.6			

**Table 3 genes-11-01479-t003:** A summary of functionally related microsatellite marker-based comparative mapping showing maximum syntenic relationships of the chromosomes of *M. acuminata* with those of foxtail millet, rice and sorghum.

Musa Chromosomes	Foxtail Millet Chromosomes	Rice Chromosomes	Sorghum Chromosomes
Chr 01	F03 (10; 42%)	R01 (13; 42%)	S01 (11; 41%)
Chr 02	F04 (3; 23%)	R01 (5; 25%)	S01 (6; 29%)
Chr 03	F09 (4; 16%)	R01 (5; 16%)	S02 (9; 27%)
Chr 04	F03 (9; 27%)	R01 (13; 46%)	S02 (7; 21%)
Chr 05	F02 (6; 24%)	R01 (5; 25%)	S06 (4; 20%)
Chr 06	F08 (5; 22%)	R03 (6; 20%)	S02 (6; 21%)
Chr 07	F02 (5; 28%)	R01 (6; 30%)	S03 (5; 24%)
Chr 08	F04 (6; 19%)	R01 (8; 27%)	S01 (6; 23%)
Chr 09	F04 (3; 21%)	R07 (4; 16%)	S01 (3; 18%)
Chr 10	F04 (6; 33%)	R02 (4; 24%)	S01 (5; 19%)
Chr 11	F09 (5; 33%)	R01 (3; 25%)	S04 (4; 27%)

Note: % represent the number of Musa microsatellite markers mapped on the chromosome of related plant species.

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
