# Peer review of "Genome-Wide Novel Genic Microsatellite Marker Resource Development and Validation for Genetic Diversity and Population Structure Analysis of Banana"

_genes, 2020, doi:10.3390/genes11121479_

Round 1

Reviewer 1 Report

Comments for the Authors:

The manuscript presents a set of novel microsatellite markers that are related to protein-coding and associated UTR genic regions, hereby called “functionally relevant SSR markers (FRSMs)”. Authors present a comprehensive characterization of a set of 35136 novel FRSMs, out of which 273 were experimentally validated. They evaluate the cross-taxon transferability, comparative mapping utilization of the markers, genotyping, genetic diversity studies and population studies potential of the markers. Finally, the authors present a free searchable database of the markers.

The article is written in a concise manner, with clear language and style. The Introduction part clearly and thoroughly describes the background and aims of the study, and Materials and Methods section is instructive and carefully describes the methodology used. Section Results and Discussion are written clearly, the main achievements are described thoroughly.

My main concern is the use of SSRs for comparative analysis of Musa and non-Musa species. I do not think that in silico mapping of SSR markers is the best way how to reveal syntenic relationships between species representing different phylogenetic families of plants (Poaceae and Musaceae). The mapping of large number of conserved genic sequences (full-length cDNAs, ESTs or assembled RNA-seq data) seems to be more appropriate to reveal synteny and collinearity, as well as comparison of the conserved gene space among such species. Would it be possible to provide comparative analysis between Musa and foxtail millet and between Musa and rice in this manner?

General comments:

  • Figures are not correctly numbered throughout the text. The same applies to Supplementary materials. Figure S6 legend is missing.
  • Line 327: add the reference to the Table with accessions list (S1)
  • Line 339-341: the number of Fis-positive and negative loci does not sum up to 15 – which is the total number of loci in question.
  • Figure 4: part (c) – columns in the graphical representation of the Structure analysis results should be labeled with accession names or codes for better interpretation and clearer picture
  • Line 359-372: The Individual clusters in the dendrogram might be discussed more in deep. Far more studies were published that could be referred to and discussed in the view of the here gathered results.
  • Line 456: Jarret (instead of Jarnet)
  • Line 515: distance (instead of distant)
  • According to Figure S5 legend – the two SSR loci used for the gel pictures to be presented in the manuscript were not among those 15 loci selected for assessment of genetic diversity and population structure. It could be considered to include gel pictures of those markers that were involved in the final 15 loci set for evaluating the use of developed markers in genetic diversity studies. These 15 loci are the most characterized in the manuscript (in terms of measurable characteristics of an SSR marker) so it would provide a better and more comprehensive overview of the actual results obtained in the wet lab experiments as well.

Author Response

Dear reviewer,

Thanks for extensive comments on the MS. According to the suggestion we thoroughly revise the MS where necessary. The point-by-point response are attach here. All the changes are marked in the revised version of MS.

Thanks.

Reviewer 2 Report

Congratulations on an interesting, worthwhile, thorough and well presented study. Making SSR markers available to the community in a well organised database is appreciated. A few minor comments on the document attached. 

Author Response

(The authors gave the same response as above.)
